# MicroRNAs: Tiny Regulators of Gene Expression with Pivotal Roles in Normal B-Cell Development and B-Cell Chronic Lymphocytic Leukemia

**DOI:** 10.3390/cancers13040593

**Published:** 2021-02-03

**Authors:** Katerina Katsaraki, Paraskevi Karousi, Pinelopi I. Artemaki, Andreas Scorilas, Vasiliki Pappa, Christos K. Kontos, Sotirios G. Papageorgiou

**Affiliations:** 1Department of Biochemistry and Molecular Biology, Faculty of Biology, National and Kapodistrian University of Athens, 15701 Athens, Greece; kkatsaraki@biol.uoa.gr (K.K.); pkarousi@biol.uoa.gr (P.K.); partemaki@biol.uoa.gr (P.I.A.); ascorilas@biol.uoa.gr (A.S.); 2Second Department of Internal Medicine and Research Unit, University General Hospital “Attikon”, 12462 Athens, Greece; vaspappa@med.uoa.gr

**Keywords:** miRNAs, normal B-cell development, B-CLL, miRNA-transcription factor network, regulation, biomarker, therapy, prognosis, diagnosis, progression, prediction

## Abstract

**Simple Summary:**

The involvement of miRNAs in physiological cellular processes has been well documented. The development of B cells, which is dictated by a miRNA-transcription factor regulatory network, suggests a typical process partly orchestrated by miRNAs. Besides their contribution in normal hematopoiesis, miRNAs have been severally reported to be implicated in hematological malignancies, a typical example of which is B-cell chronic lymphocytic leukemia (B-CLL). Numerous studies have attempted to highlight the regulatory role of miRNAs in B-CLL or establish some of them as molecular biomarkers or therapeutic targets. Thus, a critical review summarizing the current knowledge concerning the multifaceted role of miRNAs in normal B-cell development and B-CLL progression, prognosis, and therapy, is urgent. Moreover, this review aims to highlight important miRNAs in both normal B-cell development and B-CLL and discuss future perspectives concerning their regulatory potential and establishment in clinical practice.

**Abstract:**

MicroRNAs (miRNAs) represent a class of small non-coding RNAs bearing regulatory potency. The implication of miRNAs in physiological cellular processes has been well documented so far. A typical process orchestrated by miRNAs is the normal B-cell development. A stage-specific expression pattern of miRNAs has been reported in the developmental procedure, as well as interactions with transcription factors that dictate B-cell development. Besides their involvement in normal hematopoiesis, miRNAs are severally implicated in hematological malignancies, a typical paradigm of which is B-cell chronic lymphocytic leukemia (B-CLL). B-CLL is a highly heterogeneous disease characterized by the accumulation of abnormal B cells in blood, bone marrow, lymph nodes, and spleen. Therefore, timely, specific, and sensitive assessment of the malignancy is vital. Several studies have attempted to highlight the remarkable significance of miRNAs as regulators of gene expression, biomarkers for diagnosis, prognosis, progression, and therapy response prediction, as well as molecules with potential therapeutic utility. This review seeks to outline the linkage between miRNA function in normal and malignant hematopoiesis by demonstrating the main benchmarks of the implication of miRNAs in the regulation of normal B-cell development, and to summarize the key findings about their value as regulators, biomarkers, or therapeutic targets in B-CLL.

## 1. Introduction

B-cell chronic lymphocytic leukemia (B-CLL) is the most common type of leukemia in the Western World affecting mainly elders. It accounts for 7% of non-Hodgkin lymphomas and it is classified as a low-grade B-cell non-Hodgkin lymphoma. Biologically, it is characterized by the accumulation of abnormal B cells in the blood, bone marrow, lymph nodes and the spleen. Approximately 80–90% of B-CLL patients have chromosomal abnormalities, with the most common one being in the chromosomal region 13q14.3 [1]. Furthermore, numerous mutated genes have been characterized in the genome of B-CLL patients which occur mainly in the *NOTCH1*, *MYD88*, *TP53*, *ATM*, *SF3B1* and *BIRC3* genes. B-CLL is a highly heterogeneous disease and therefore, timely, specific, and sensitive assessment of this malignancy and its progression is vital. Therefore, in the last decades, several molecular and clinical prognostic markers have been proposed for B-CLL assessment. Age, Binet and Rai staging systems, deletions in chromosomes 11q, 13q, 17p, serum markers such as β2-microglobulin, cell surface markers such as CD38, *IGHV* mutational status, z-associated protein-70 (ZAP70) and lipoprotein lipase (LPL) represent some of these markers [2].

MicroRNAs (miRNAs) are small non-coding RNAs (sncRNAs), with an average length of 22 nucleotides. These sncRNAs are transcribed and processed by Drosha and Dicer enzymes and target mRNAs post-transcriptionally via recruitment of Argonaute (AGO) proteins and therefore, recruitment of the relative RISC complexes to a complementary mRNA target sequence. This recruitment leads to mRNA degradation and/or repression of translation. Furthermore, miRNAs appear as a complex regulatory network as multiple miRNAs target the same mRNA and multiple mRNA sequences are targeted by one specific miRNA. During the last few years, the role of miRNAs has been evaluated in various diseases including cancer. These studies highlighted the remarkable significance of miRNAs as regulators, biomarkers for diagnosis, prognosis, progression and prediction, as well as molecules with therapeutic utility [3]. Unique miRNAs and other factors contributing to B-CLL pathogenesis are summarized in Figure 1.

Until today, various studies delineate the regulatory role of miRNAs in hematological malignancies, as they are critically implicated in B-cell development, proliferation and migration, by affecting numerous pathways including the BCR signaling, the mitogen-activated protein kinase (MAPK/ERK), the phosphatidylinositol 3–kinase (PI3K)/serine/threonine kinase AKT, and the nuclear factor κB (NFkB). Since 2002, when Calin et al. reported a downregulation in the expression levels of miR-15a-5p and miR-16-5p as a result of the deletion located at the chromosome 13q14 [4], which is the most common among B-CLL patients, great progress has been made in order to reveal the role of miRNAs in B-CLL. At present, miRNAs have been characterized as valuable regulators and biomarkers for the assessment of the occurrence and progression of B-CLL as well as evaluators of therapy and promising molecules for therapeutic strategies.

Even though the implication of miRNAs in normal B-cell hematopoiesis, lymphomas and leukemias has been extensively studied [5], scientific knowledge in this context remains limited with only specific miRNAs being studied broadly. Moreover, a review summarizing and highlighting their significant involvement in normal B-cell development and B-CLL is still missing. Therefore, in this review, we aim to highlight the importance of miRNAs in normal B-cell development both in the bone marrow and the periphery and delineate their roles in B-CLL regulation, therapy and disease assessment. Moreover, we propose promising topics for future, targeted research concerning the regulatory potential of miRNAs and their establishment in clinical practice.

## 2. miRNAs: How Are They Involved in Normal B-Cell Development?

The development of B cells is a multi-step and tightly regulated process. In brief, the developmental procedure consists of the following stages: the developmental phases, that take place in the bone marrow, include the transformation of hematopoietic stem cells (HSCs) to common lymphoid progenitors, pro-B cells, pre-B cells, and immature B cells. These developmental stages are characterized by rearrangements in the Ig genomic loci, called VDJ recombination. This process leads to the formation of Ig heavy chains at the pro-B stage and Ig light chains at the pre-B-cell stage, which compose a transmembrane protein called BCR. Next, the BCR-expressing immature B cells undergo the central tolerance checkpoint, during which those expressing a self-reactive antibody are eliminated. Following, B cells migrate in the spleen where naïve B cells bind to an antigen and differentiate into follicular or marginal zone B cells. This differentiation strongly depends on BCR signals. Next, marginal zone B cells populate the marginal zone, while follicular B cells enter the germinal center, forming three distinct zones: the dark, light, and mantle zone. Subsequently, germinal center B cells differentiate into memory or plasma cells [6].

### 2.1. miRNAs in Bone Marrow B-Cell Development

miRNAs have been multifariously implicated in the regulation of B-cell development, affecting most of the stages composing this process. The global impact of miRNAs is demonstrated by the fact that the lack of DGCR8, a key molecule for proper miRNA biogenesis, led to elevated apoptosis rate of early B cells [7,8]. Moreover, the expression of each miRNA during B-cell development seems to be stage-specific, as it was uncovered by the study of Spierings et al. [9]. Other studies support this finding as well; miR-181a-5p, miR-150-5p, miR-132-3p, and miR-126-3p were differentially expressed among developmental stages [10,11]; moreover, the ectopic overexpression of miR-181a-5p in common lymphoid progenitors led to an increase in the total number of B cells, indicating its involvement in B-cell development [12]. In contrast, overexpression of a member of the miR-23a cluster, miR-23a-5p, in HSCs led to a decrease in total B-cell number [13].

Besides their involvement in the general context of B-cell development, several miRNAs have been demonstrated to affect specific early stages of the developmental process by interacting with transcription factors crucial for normal B-cell development. The main transcription factors that orchestrate the development of B cells are TCF3, EBF1, and PAX5 [14,15]. TCF3 is required for the initiation and maintenance of the developmental procedure, as well as for the recruitment of EBF1 and PAX5 [16]; both these transcriptional factors are essential for early B-cell differentiation, as they participate in the formation of a functional BCR [15,17]. Interestingly, in absence of EBF1, the developmental process is not abolished, since miR-126-3p was shown to partly rescue B-cell development, by inducing the expression of genes that are required for the process, including *RAG1* and *RAG2*, two recombinases responsible for VDJ recombination [11,18]; this fact comes in line with its overexpression in early developmental stages. Other transcription factors are implicated too, including FOXP1, which controls RAG1 and RAG2 expression [19], and EGR1, a transcriptional regulator essential for B-cell differentiation [20]. In this context, miR-191-5p was shown to target *Foxp1*, *Tcf3*, and *Egr1* in mice, exerting a potential controversial role in the developmental procedure since both its deletion and overexpression led to inefficient development of the B cells [21]. *Egr1* is also decreased upon miR-146a-5p overexpression in mice, leading to B-cell malignancies [20].

Such a regulatory network, consisting of miRNAs and transcription factors, has been described to arrest the developmental procedure at the pro- to pre-B-cell transition as well; this transition is an important checkpoint in the B-cell development process. Overexpression of miR-150-5p in common lymphoid progenitors, blocked pro- to pre-B-cell transition, due to its binding to *MYB*, a transcription factor whose deletion completely abolishes B-cell development; this fact explains the above-mentioned stage-specific expression of miR-150-5p, as normally it is not detected in early developmental stages [10,22,23]. Similar to miR-150-5p, miR-132-3p overexpression in mouse HSCs blocked the developmental procedure at the pro-B-cell stage by targeting another transcription factor, namely *Sox4*, which is required for the survival of pro-B cells and is involved in Rag1 expression; the expression of miR-132-3p is stage-specific as well and is likely to be induced by BCR signaling, which means that it is normally expressed after the transition stage [24]. These typical examples indicate that the stage-specific expression pattern is not stochastic but strictly linked to each miRNA utility and function in the developmental process.

Additionally, miR-24-3p, which belongs to the previously mentioned miR-23a cluster, had similar results by targeting the transcription factor *MYC* [25]; this information can explain the B-cell development blocking, caused by the miR-23a cluster [10]. Additionally, MYC controls the expression of the miR-17/92 cluster; its overexpression arrests B cells at the pro-B-cell stage, by inhibition of the proapoptotic protein BCL2L11, and of PTEN, which is implicated in the vital PI3K signaling pathway [26,27]. This axis is also implicated in immature B cells; the targeting of *PTEN* and *BCL2L11* by this cluster and by miR-148a-3p promotes the survival of immature B cells, as well as the production of self-reactive antibodies, and thus leads to their elimination at the central tolerance checkpoint [28,29,30]. All this information highlights the great and multifaceted impact of miRNAs in early-stage B-cell development.

### 2.2. miRNAs in Peripheral B-Cell Development

Concerning the development in the periphery, several miRNAs have been reported to play a role, too, with most of them affecting germinal center B to plasma cell transition. miR-155-5p is essential for plasma cell production, as it inhibits SPI1 proto-oncogene, leading to downregulation of PAX5 [31]; PAX5 downregulation is essential for terminal B-cell differentiation. Moreover, miR-148a-3p promotes germinal center to plasma cell transition, by inhibiting *BACH2* and *MITF*; these transcription factors repress the transcription factors *PRDM1* and *IRF4*, both involved in premature plasma cell differentiation, by initiating cascades of gene expression changes and inducing class-switch recombination [32], a process of further recombination of the Ig genes, essential for terminal B-cell differentiation [6]. On the contrary, miR-125b-5p was shown to negatively regulate the B-cell differentiation in germinal centers, through targeting the transcription factors *PRDM1* and *IRF4* [33], thus its physiological silencing is required for normal B-cell development [34]. This fact is also supported by a study that demonstrated that transgenic mice overexpressing miR-125b-5p developed lethal B-cell malignancies [35]. *PRDM1* was also shown to be targeted by two members of the miR-30 family, namely miR-30b-5p and miR-30d-5p, and miR-9-5p, leading to similar results [36].

Besides germinal center B cells, miRNAs were shown to affect marginal zone B cells, too, which are involved in the early rapid response to infection. miR-146a-5p overexpression was demonstrated to reduce the total number of marginal zone B cells, by targeting *NUMB* which prevents TP53 degradation and upregulates the Notch signaling pathway; this pathway promotes the development of marginal zone lymphocytes [37,38,39]. Aberrant development of marginal zone B cells is also a consequence of miR-142-5p. Moreover, its deficiency in nude mice led to a deregulation of gene expression in mature B cells, due to higher levels of B cell-activating factor receptor (BAFFR; also known as TNFRSF13C), which is a direct target of miR-142-5p and enhances B-cell survival, leading to robust B-cell proliferation [40].

All this information, as well as additional miRNAs participating in B-cell development, are summarized in Table 1; those miRNAs that were shown to have a great impact in the developmental process are presented in Figure 2.

## 3. miRNAs: Regulators, Biomarkers and Potential Therapeutic Entities in B-CLL

### 3.1. miRNAs as Regulators in B-CLL

miRNAs appear as important molecules in the regulation of B-CLL either by directly regulating key factors that are involved in B-CLL pathogenesis or after the alteration of their levels by epigenetic modifications. They can act as oncogenic molecules or as tumor suppressors with a remarkable involvement in several B-CLL signaling pathways. They appear as crucial regulators of cellular procedures such as early B-cell development [45], cell metabolism and autophagy [46,47], of signaling pathways including the NFkB pathway [20,48], the Hedgehog (Hh) signaling pathway [49], as well as of key molecules in B-CLL such as BTK [50], TCL1A [51,52,53], BCL2 [54], TERT [55], and the heat shock proteins (HSPs) [56]. Moreover, they have an exceptional involvement in distinct parts of the BCR signaling pathway, which is abnormally activated in B-CLL, as well as, in its downstream pathways [57]. This interaction leads to the regulation of apoptosis, survival, proliferation and migration of leukemic B cells.

Initially, a differential expression of specific miRNAs was found between BCR stimulated and unstimulated cells [58]. Interestingly, in leukemic B cells, high expression of miR-155-5p mimics led to an increased BCR stimulation compared to controls. On the contrary, miR-155-5p inhibition resulted in a reduced calcium flux which is a result of BCR stimulation reduction [59]. In another study, miR-150-5p was found to regulate BCR signaling in B-CLL by regulating the expression levels of *GAB1* and *FOXP1*. GAB1 is an adaptor molecule that recruits numerous factors, including PI3K, enhancing the BCR signaling. FOXP1 is a transcription factor that is strongly expressed after B-cell activation. Characteristically, transfection of cells with miR-150-5p mimics decreased the levels of *GAB1* and *FOXP1*. Moreover, sensitive B-CLL samples to BCR stimulation had higher levels of *GAB1* and *FOXP1* mRNAs. Additionally, insensitive to BCR stimulation samples had high levels of miR-150-5p, reinforcing the idea of the connection between high miR-150-5p levels with low *GAB1* and *FOXP1* expression which results in a reduced sensitivity [60]. Additionally, FOXP1 was also found to be controlled by miR-34a-5p with its levels being reduced during DNA damage response leading to a limitation of BCR signaling [61]. Furthermore, a recent study revealed the involvement of miR-29 family in CD40 signaling by targeting *TRAF4* and proposed a novel regulatory axis in Β-CLL, modulated by the BCR activity [62].

Another study revealed the involvement of miR-21-5p in BCR-mediated MAPK/ERK signaling. miR-21-5p was found to downregulate *SPRY2* expression, leading to a decrease of SPRY2 levels in MEC-1 cells, a human B-CLL cell line. SPRY2 is an inhibitory protein that interacts with RAF1, BRAF, and SYK for the downregulation of the MAPK/ERK signaling in leukemic B cells. Therefore, high expression of miR-21-5p leads to an upregulation of MAPK/ERK signaling and a relative result to survival and proliferation of the leukemic B cells [63]. However, tumor-suppressive miRNAs which contribute negatively to the MAPK/ERK signaling pathway have also been identified in B-CLL [64,65]. As described previously, miRNAs are also involved in the regulation of the PI3K/AKT pathway, either by directly influencing the pathway or by downregulating the expression of *PTEN,* a tumor suppressor of this pathway [66,67]. This pathway regulates cellular growth, metabolism and survival.

Another important example of miRNAs regulation in B-CLL is the connection of miR-15a-5p and miR-16-5p with TP53 and both miR-34b-3p and miR-34c-5p–produced by genes located in chromosomes 13q, 17p, and 11q, respectively–and their relation to B-CLL pathogenesis and patients’ outcome. The deletion of 13q14 in B-CLL patients results to downregulation of miR-15a-5p and miR-16-5p levels and high levels of its targets, BCL2 and MCL1 anti-apoptotic proteins, leading to a reduction of apoptosis. Furthermore, low miR-15a-5p and miR-16-5p levels, caused by 13q14 deletion, results in an upregulation of their target, TP53, which leads to increased levels of miR-34b-3p and miR-34c-5p, thus leading to reduction of ZAP-70 levels and its downstream pathways, as well as to an indolent B-CLL phenotype [68].

Epigenetic modifications of miRNAs may lead to significant alterations in their expression and function. Baer et al. found a negative correlation between the DNA methylation status of miRNA promoters and the expression of their corresponding mRNA targets [69], proposing a strong epigenetic effect in the miRNA expression after a methylation change in the DNA promoter region. Two other studies highlighted the significance of methylation in the *MIR34B*/*MIR34C* promoter region. The methylation status of the *MIR34B*/*MIR34C* promoter region differed between normal and B-CLL cohorts [70,71], proposing a correlation between *MIR34B*/*MIR34C* promoter methylation, the downregulation of mir-34b/mir-34c and the respective mature miRNAs, and subsequently the reduction of their tumor-suppressive activity in B-CLL patients. Furthermore, acetylation alteration may differentiate the levels of specific miRNAs as it was found that histone deacetylases may mediate the silencing of miR-15a-5p, miR-16-5p, and miR-29b-3p in B-CLL [72]. This alteration is of high importance as these miRNAs are significant regulators in B-CLL. Furthermore, it is important to mention that epigenetic alterations in specific miRNAs, such as RNA editing, which were changes of adenosine to inosine and cytosine to uracil, have been found altered between leukemic and normal B cells and may differentiate the targets of these miRNAs [73].

Recent studies have also examined the interaction of miRNAs with a relatively new RNA type, circular RNAs (circRNAs), in the context of B-CLL pathogenesis. CircRNAs, which are mainly produced by backsplicing events in pre-mRNAs, bear a circular structure and can regulate cell function, multifariously, including sponging of miRNAs, which consequently limits the effect of miRNAs. A relative example is circ-CBFB, which was predicted to sponge miR-607, leading to an upregulated expression of the miRNA target, *FZD3*, which is a receptor for WNT proteins. This upregulation leads to the activation of the WNT/β-catenin pathway and B-CLL progression [74]. Additionally, the downregulation of circ_0132266, which acts as a sponge for miR-337-3p, resulted in a downregulation of *PML*, which is the target molecule of the latter and a tumor suppressor. The downregulation of PML leads to increased cell viability [75].

All the aforementioned information delineates a great involvement of miRNAs in B-CLL regulation as they can modulate oncosuppressors and oncogenes, possess a key role in a plethora of signaling pathways and can also be epigenetically regulated having different functions. Specific miRNAs with a regulatory role in B-CLL are summarized in Table 2.

### 3.2. miRNAs as Diagnostic, Prognostic and Predictive Biomarkers in B-CLL

In 2004, Calin et al. found deregulation in the expression levels of numerous miRNAs between leukemic B cells and normal CD5^+^ B cells, as well as, among distinct molecular B-CLL subtypes [82]. This study indicated for the first time a potential value of miRNAs as diagnostic and prognostic biomarkers for B-CLL. One year later, the authors proposed the first miRNA-signature for prognosis and progression of this malignancy which could be used in order to distinguish between *IGHV*-mutated/ZAP70-positive and *IGHV*-unmutated/ZAP70-negative patients [83].

Today, many scientists propose miRNAs as ideal biomarkers as they are highly stable and can be easily detected and quantified in blood, other body fluids and fresh or paraffin-embedded tissues. Over the years, numerous other miRNAs have been proposed in B-CLL as biomarkers for risk assessment, prognosis, prediction and progression. In the last few years, research has focused on assessing the value of specific miRNAs for the disease rather than a panel of numerous miRNAs. A typical example is miR-181b-5p, which has been identified as a biomarker of progression from indolent to aggressive B-CLL [84]. Moreover, miR-155-5p expression levels were elevated in individuals with monoclonal B-cell lymphocytosis compared to normal blood donors and in patients with B-CLL compared to patients with monoclonal B-cell lymphocytosis [85]. Furthermore, miR-155-5p levels were characterized as a valuable prognostic biomarker and biomarker for the risk assessment of B-CLL development [86]. Moreover, it is worth mentioning that other miRNAs appear as promising biomarkers in distinct subgroups of B-CLL patients with different cytogenetic abnormalities [87,88,89].

Based on the high potential of miRNAs as biomarkers, there are studies, which incorporate miRNAs in prognostic models, which provide B-CLL–specific prognostic scores. In this context, Stamatopoulos et al. proposed a molecular prognostic score for the disease which includes the expression levels of miR-29c-3p, ZAP-70 and LPL for the stratification of B-CLL patients into three distinct groups concerning treatment-free and overall survival (OS) prognosis [90]. Another system, the 21FK score, has been proposed for prognosis assessment of B-CLL patients. This score examines the expression levels of miR-21-5p by qRT-PCR, chromosomal abnormalities with fluorescence in situ hybridization, and karyotype in peripheral blood mononuclear cells (PBMCs) of B-CLL patients, in order to stratify patients for OS prognosis assessment [91]. Patients with a high 21FK score had a shorter OS time and therefore worse prognosis. Even though these scoring systems could provide a satisfactory prognosis, they have not been adopted in clinical practice.

Several studies revealed the predictive biomarker utility of miRNAs in B-CLL. These studies assess whether the treatment strategy will be effective in patients with distinct characteristics. For instance, particular miRNAs have emerged as predictive biomarkers in order to distinguish fludarabine resistant or rituximab resistant patients from responsive individuals for each therapeutic strategy [92,93]. Moreover, it is worth mentioning a study that found that miR-34a-5p low expression was associated with TP53 inactivation, which is a tumor suppressor with a key role in the induction of apoptosis of leukemic B cells, regardless of 17p deletion or *TP53* mutation. This inactivation of TP53 was observed independently of the mutational status of the *TP53* gene or deletions in 17p chromosomal region, where the *TP53* gene is located. Consequently, low levels of miR-34a-5p denote apoptosis resistance and fludarabine refractory disease [94]. Additionally, the expression levels of miR-21-5p, miR-148a-3p and miR-222-3p could also serve for the discrimination of fludarabine-refractory B-CLL patients from fludarabine-sensitive ones. Inhibition of miR-21-5p and miR-222-3p was found to increase caspase activity in fludarabine-treated *TP53*-mutant MEG-01 chronic myelogenous leukemia cells, suggesting these two miRNAs as key factors of acquisition of resistance to fludarabine [95]. Another noteworthy study noticed an inversive correlation between circulating miR-125b-5p and miR-532-3p expression levels, rituximab-induced lymphodepletion and CD20 expression on CD19+ T cells in patients with B-CLL [93]. All the aforementioned information highlights the high and complicated importance of miRNAs for the risk assessment of B-CLL development, progression and treatment prediction. Unique miRNAs, miRNA signatures and scoring systems, which have been proposed for the assessment of B-CLL are summarized in Table 3.

### 3.3. miRNAs in B-CLL Therapy

The treatment of B-CLL consists of numerous therapeutic strategies and involves alkylating agents, glucocorticoids, purine analogs, monoclonal antibodies and bone marrow transplantation. Moreover, target-specific therapies have also emerged, targeting BCR receptor, BTK, PI3K and apoptosis-related proteins. Taking into consideration the multifaceted roles of miRNAs in B-CLL regulation, the fact that they are naturally produced molecules by organisms and that their levels may be easily regulated with miRNA-mimics or miRNA-antagomiRs, miRNAs appear as promising therapeutic molecules and therapeutic targets for this disease.

Specific studies emphasized the therapeutic potential of miRNAs. Interestingly, Salerno et al. described an increase of drug sensitivity in the New Zealand Black (NZB) mouse cell line, LNC, after the correction of the miR-15a-5p and miR-16-5p defect. NZB mouse model has a genetically determined age-associated increase in malignant B-1 clones and decreased expression of miR-15a-5p and miR-16-5p in B-1 cells. A cell cycle arrest in the G_1_ phase was observed after the exogenous addition of miR-16-5p mimics. This observation is in correlation with the decrease of *CCND1* levels–an overexpressed gene in some human B-CLL cases–as miR-15a-5p and miR-16-5p target the 3′ UTR of *CCND1*. Moreover, a synergetic effect of miR-16-5p and chemotherapeutic agents was observed in the induction of apoptosis [102]. In another study where lentiviral vectors were used for the in vivo restoration of miR-15a-5p and miR-16-5p in NZB mouse model, mice appeared with a moderate B-CLL phenotype. The lentivirus delivering system assisted in low systemic toxicity levels and limited off-target effects. These results came in line with those of another study, which demonstrated that the effect of the restoration of these two miRNAs were the increased expression of miR-15a-5p and miR-16-5p both in transduced cells and serum and the decreased viability of B-1 cells [103].

Numerous other miRNAs have also been associated as regulators of B-CLL therapy [104,105], with miR-181a-5p and miR-181b-5p constituting promising examples. Leukemic B cells from *TP53*^wt^ patients were transfected with miR-181a-5p and miR-181b-5p mimics resulting in a significant increase in apoptosis compared to controls, with no effect being observed in B-CLL patients with a decreased expression of *TP53* [106]. Moreover, miR-181b-5p was found to affect the levels of TCL1A, AKT, and both phosphorylated ERK1 and ERK2, to reduce leukemic cell expansion and to increase survival of a treated transgenic mouse model [107].

miRNAs have also been characterized as oncogenic for numerous malignancies leading to a required downregulation of these miRNAs for the improvement of the cancer patients’ outcome. In B-CLL, ibrutinib suppresses the expression of oncogenic miRNAs, leading to a downregulation of malignant B-cell proliferation [108]. Moreover, in another study, the downregulation of miR-17-5p expression levels was proposed as a potential therapeutic strategy for B-CLL. An in-vitro administration of antagomiR-17-5p, which is a miRNA inhibitory oligonucleotide molecule, in MEC-1 cells significantly reduced miR-17-5p expression levels and cell proliferation. Moreover, tumors generated by MEC-1 cells injected into severe combined immunodeficiency mice, which were treated with antagomiR-17-5p, presented an inhibition of growth and complete remission in 20% of the cases. Furthermore, antagomiR-17-5p treated mice possessed a longer median OS in comparison to the controls, while no signs of toxicity were observed [109].

Besides in vivo lentiviral delivery of miRNA mimics as a therapeutic strategy for B-CLL, antibody-based strategies have also been proposed for the delivery of miRNA mimics and antagomiRs. These strategies include the construction of particles which include the selected miRNAs and are conjugated with antigen-specific antibodies for characteristic markers of leukemic B cells, such as CD38 and ROR1. Therefore, these “vehicles” are attached to leukemic B cells which are expressing these markers [110,111]. The findings of studies proposing miRNAs as molecules for therapeutic strategies are summarized in Table 4.

The use of miRNA mimics or antagomiRs has entered clinical trials with promising results for targeted therapy of various diseases. Therefore, these molecules appear as promising therapeutic agents for the future as they could expand even more the field of personalized medicine. However, for miRNA mimics or antagomiRs to succeed, off-target phenomena and severe effects should be eliminated, and delivery strategies should be improved. However, prior to all these ameliorations, it is important to clarify the miRNA regulatory disease-specific network of miRNAs with deregulated levels in B-CLL, in order to distinguish the most promising miRNAs for treatment, and therefore achieve a miRNA-based therapy.

### 3.4. Viral miRNAs in B-CLL

Epstein-Barr virus (EBV) is a ubiquitous oncogenic human herpesvirus implicated in lymphomas, such as Burkitt’s lymphoma, while recent studies have associated EBV infection with Β-CLL progression [113]. More specifically, it was shown that Β-CLL patients with elevated levels of EBV DNA load had significantly shorter OS time intervals and were characterized by therapy resistance, compared to the ones with lower levels of EBV DNA load [114,115]. However, further investigation is required regarding the potential mechanisms of EBV-driven oncogenesis in Β-CLL patients.

A proposed mode of action, via which EBV sustains viral infection, evades host immunity and potentially leads to oncogenesis, is based on viral miRNAs [116]. EBV was the first virus shown to encode viral miRNAs. To date, 44 mature miRNAs that could be classified into two clusters (*BHRF1* cluster and *BART* cluster) have been identified in EBV. Particularly, the miRNAs from the *BHRF1* cluster are expressed during lytic infection, inhibit apoptosis, and favor proliferation of infected cells to enable the early phase of viral propagation. Considering the role of these viral miRNAs, miR-BHRF1-1 has been further investigated in the context of Β-CLL. Precisely, the expression levels of miR-BHRF1-1 were significantly higher in the plasma of Β-CLL patients compared to the plasma of healthy individuals, while elevated levels of this miRNA were positively associated with advanced Rai stages. Furthermore, Β-CLL patients with elevated expression levels of miR-BHRF1-1 were characterized by shorter OS time intervals, compared to the ones with lower expression levels [117]. These findings designate the potential value of this miRNA as biomarker. Additionally, high expression levels of miR-BHRF1-1 were positively correlated with high miR-155-5p expression levels. This observation is quite significant, since miR-155-5p plays a decisive role in both normal B-cell development and Β-CLL progression, as it has been thoroughly analyzed above. The association between these two miRNAs has been also observed in another study, which showed that cellular miR-155-5p was induced by the viral miRNA and that miR-155-5p played a key role in B-cell immortalization [118]. Finally, infection of leukemic B cells with miR-BHRF1-1 reduces the levels of the key tumor suppressor, TP53 [117]. This interaction is, also, supported by an independent study, which suggested that this miRNA exerts its role in B-CLL via downregulation of TP53 and uncovered the therapeutic potential of miR-BHRF1-1 [119].

All these findings advocate the implication of EBV miRNAs in B-CLL onset and progression. Even though this research field is still in its infancy, the elucidation of the role of viral miRNAs in B-CLL progression is quite promising since it could further assist the discovery of novel biomarkers and therapeutic strategies.

## 4. Future Perspectives

Although extensive research has been conducted, only a few miRNAs appear as important regulatory molecules both in normal B-cell development and B-CLL, as depicted in Figure 3. These include members of the miR-17/92 cluster, miR-21-5p, miR-29 family, miR-34 family, miR-125b-5p, miR-150-5p, miR-155-5p, and miR-181 family. In particular, miR-34a-5p was found to act similarly in both situations targeting *FOXP1*, either by blocking the development of B cells in normal development or by limiting proliferation and survival of leukemic B cells. miR-150-5p appears to function in a similar way, as well. On the contrary, miR-181 isomiRs possess an opposite function, as they promote normal B-cell development, but reduce leukemic cell expansion in B-CLL.

The implication of the aforementioned miRNAs in normal B-cell development indicates that fully unraveling the regulatory network that orchestrates the typical developmental pathway of B cells can gain new insights in B-CLL understanding. Even though a great number of studies delineate the roles of miRNAs in normal B-cell development and B-CLL, there is no clear evidence of a large regulatory network of miRNAs in these two different conditions. Moreover, no safe conclusions can be drawn, as scientific knowledge seems to focus either on specific miRNAs that could regulate for instance a specific pathway or on miRNA signatures with significant differences in their expression levels between different stages of the developmental process or distinct B-CLL conditions. Additionally, some findings seem contradictory such as the paradigm of cellular and serum circulating miR-150-5p that was found to possess an opposite prognostic significance in patients with B-CLL [120]. Moreover, specific miRNAs such as miR-125a-5p and miR-34a-5p which were proposed for the prediction of Richter syndrome, a lethal complication in B-CLL patients [121], may appear as ideal molecules for further research in order to distinguish molecular pathways that contribute to distinct subgroups of B-CLL patients. Taking into consideration the significant presence of miRNAs both in normal B-cell development and B-CLL, further research is required to shed light on the involvement of miRNAs in normal B-cell development and the pathogenetic events that lead to B-CLL.

Another important aspect is that polymorphisms in pre-miRNAs and genes which are involved in miRNAs biogenesis pathway were found to contribute to the risk of B-CLL [122]. miRNAs can be epigenetically regulated by multiple processes and therefore possess a different regulatory effect. These epigenetic regulations may remarkably enlarge the regulatory potential of miRNAs and appear as another promising area for research.

Increasing evidence shows specific miRNAs with deregulated levels in Β-CLL, in comparison with normal individuals, suggesting miRNAs as biomarkers. Moreover, the alteration of their levels is a tightly regulated and finely tuned process. Therefore, elucidating the regulatory effect of miRNAs with deregulated levels may reveal other miRNAs with therapeutic potency. Furthermore, specific miRNAs have already been identified as molecules with significant therapeutic utility. Their impact may be extremely targeted as they regulate the expression of their specific targets and therefore, miRNAs may appear as ideal agents for combinational therapy. Lastly, therapeutic miRNA-based strategies have entered clinical trials for numerous diseases. Therefore, more targeted research is required in order to clarify the specific therapeutic potential of miRNAs in B-CLL.

## 5. Conclusions

miRNAs are undoubtedly implicated in numerous stages of the normal B-cell development either by blocking the development or by facilitating it. In B-CLL, they act as oncogenes or as oncosuppressors with key involvement in signaling pathway regulation. Moreover, they can be epigenetically regulated which can potentially lead to other regulatory effects. In addition, they are valuable biomarkers for diagnosis, prognosis and prediction for B-CLL patients and appear as promising molecules for therapeutic strategies. Even though the multifaceted role of miRNAs in B-CLL has been extensively studied in the last few years, important information is still missing, while no molecule has emerged as a validated regulator in numerous pathogenetic pathways of this malignancy. As a result, their regulatory potency accompanied by their therapeutic ability are two topics that require further targeted research.

## Figures and Tables

**Figure 1 cancers-13-00593-f001:**
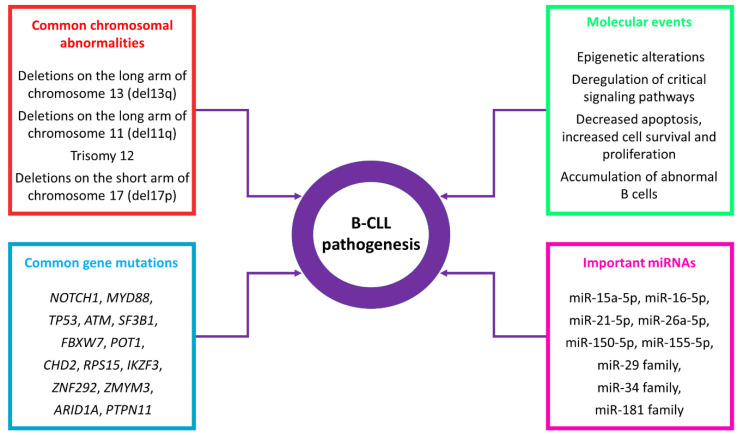
Common genetic alterations, molecular events, and miRNAs contributing to B-CLL pathogenesis.

**Figure 2 cancers-13-00593-f002:**
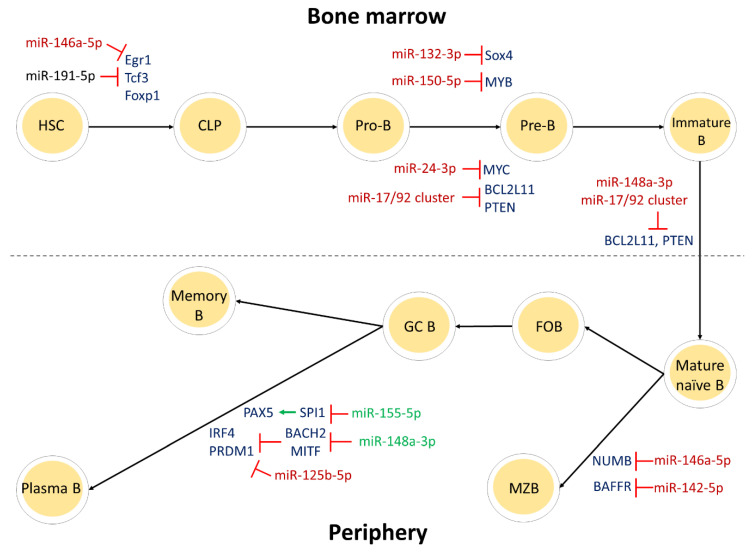
The main steps of B-cell development, and some of the miRNAs implicated in this process. Red color in lines and miRNA red font color indicate inhibition of expression, while green color in arrows and miRNA font indicates induction of expression. Black font color indicates a miRNA acting as a rheostat for the developmental process. CLP, common lymphoid progenitor; FOB, follicular B cell; MZB marginal zone B cell; GC B, germinal center B cell.

**Figure 3 cancers-13-00593-f003:**
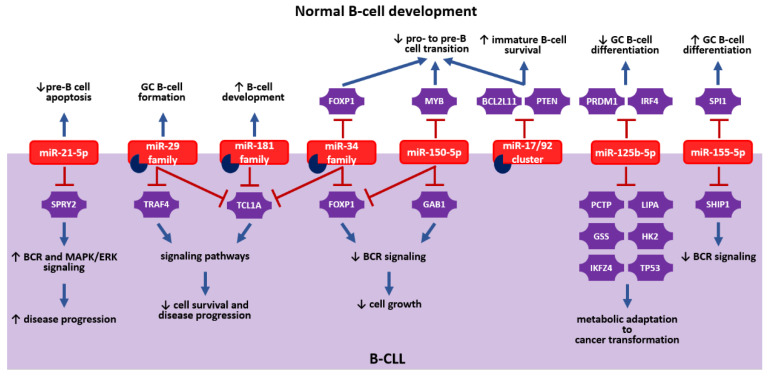
The regulatory roles and specific targets of miRNAs, which are involved in normal B-cell development and B-CLL. Red inhibitory signs indicate the downregulation of specific targets by miRNAs and blue arrows indicate the regulatory impact of the miRNAs. Partial circles indicate the potential therapeutic utility of the miRNAs. miRNAs can promote (↑) or suppress (↓) numerous biological processes. With regard to the miR-17/92 cluster, only miR-17-5p has been proposed as a therapeutic target for B-CLL. GC B, germinal center B cell.

**Table 1 cancers-13-00593-t001:** miRNAs implicated in normal B-cell development.

	miRNA	Target	Effect	References
Bone marrow	miR-181a-5p	-	Promotes B-cell development	[12]
	miR-23a-5p	-	Inhibits B-cell development	[13]
	miR-191-5p	Tcf3, Foxp1, and Egr1	Acts as a rheostat for early B-cell development in mice	[21]
	miR-126-3p	RAG1 and RAG2	Rescues B-cell development in absence of EBF1	[11]
	miR-146a-5p	Egr1	Downregulates Egr1, leading to B-cell malignancies	[20]
	miR-150-5p	MYB	Inhibits pro- to pre-B-cell transition	[10,22]
	miR-132-3p	Sox4		[24]
	miR-34a-5p	FOXP1		[41]
	miR-24-3p	MYC		[13,25]
	miR-17/92 cluster	BCL2L11 and PTEN	Inhibits pro- to pre-B-cell transition; promotes the survival of immature B cells	[26]
	miR-21-5p	-	Inhibits pre-B cell apoptosis	[42]
	miR-148a-3p	BCL2L11, PTEN, and GADD45A	Promotes the survival of immature B cells	[28]
	miR-210-3p	-	Inhibits autoantibody production in mice	[43]
Periphery	miR-29a-3p	-	Is essential for the formation of germinal center B cells in mice	[44]
	miR-125b-5p	PRDM1 and IRF4	Inhibits the differentiation of germinal center B cells	[33,34,35]
	miR-9-5p	PRDM1		[36]
	miR-30b-5p/miR-30d-5p			
	miR-223-3p	LMO2		[36]
	miR-155-5p	SPI1	Promotes the differentiation of germinal center B cells	[31]
	miR-148a-3p	BACH2 and MITF		[32]
	miR-146a-5p	NUMB	Inhibits the formation of marginal zone B cells	[38]
	miR-142-5p	BAFFR	Inhibits the formation of marginal zone B cells; regulates gene expression in mature B cells	[40]

**Table 2 cancers-13-00593-t002:** miRNAs with a regulatory effect in B-CLL.

miRNA	Target	Effect	References
miR-29 family	TRAF4	Suppresses CD40 signaling	[62]
miR-202-3p	SUFU	Regulates Hedgehog (Hh) signaling	[49]
miR-607	FZD3	Enhances WNT/β-catenin signaling	[74]
miR-708-5p	IKBKB	Suppresses NFkB signaling	[48]
miR-22-3p	PTEN, CDKN1B, and BIRC5	Enhances PI3K/AKT signaling	[66]
miR-3151-5p	MADD and PIK3R2	Suppresses MAPK/ERK and PI3K/AKT signaling	[64]
miR-126-3p	PIK3R2	Suppresses MAPK/ERK signaling	[65]
miR-21-5p	SPRY2	Enhances BCR and MAPK/ERK signaling	[63]
miR-150-5p	GAB1 and FOXP1	Suppresses BCR signaling	[58,60]
miR-34a-5p	FOXP1		[61]
miR-155-5p	SHIP1	Enhances BCR signaling	[59]
	-	Regulates cell survival	[76]
miR-221-3p; miR-222-3p	CDKN1B	Regulates cell proliferation	[77]
miR-15a-5p; miR-16-5p	BCL2	Regulates cell survival	[54]
	TP53	Regulates cell proliferation	[55,68]
	ROR1	Regulates cell survival and proliferation	[78]
miR-26a-5p	EZH2	Regulates cell survival and proliferation	[79]
	PTEN		[67]
miR-214-3p	PTEN		[67]
miR-337-3p	PML		[75]
miR-106b-5p	ITCH		[80]
miR-28-5p	NDRG2		[81]
miR-650			
miR-181a-5p/miR-181b-5p	-	Suppresses cell growth	[58]
miR-210-3p; miR-425-5p; miR-1253; miR-4269; miR-4667-3p	BTK	Promotes apoptosis	[50]
miR-130a-3p	ATG2B and DICER1	Inhibits autophagy and regulates cell survival	[46]
miR-125b-5p	PCTP, LIPA, GSS, HK2, IKZF4, and TP53	Regulates metabolic adaptation to cancer transformation	[47]
miR-29b-3p; miR-34b-5p; miR-181b-5p; miR-484	TCL1A	Regulates multiple signaling pathways and cell survival	[51,52,53]

**Table 3 cancers-13-00593-t003:** miRNAs as candidate biomarkers in B-CLL.

miRNA	Localization	miRNA Expression	Biomarker Utility	References
miR-20b-5p	PBMCs ^1^	Lower levels in patients with poor prognosis	Prognosis	[96]
miR-21-5p; miR-125b-5p; miR-148a-3p; miR-181a-5p; miR-221-3p; miR-222-3p; miR-532-3p	PBMCs ^1^	Lower levels in responsive patients	Prediction of response	[92,93,95]
miR-29a-3p; miR-34a-5p		Higher levels in responsive patients		[92,94]
miR-181b-5p	PBMCs ^1^	Higher levels in indolent vs. aggressive disease	Prediction of progression	[84]
miR-744-5p		Lower levels in patients with shorter time to first treatment		[97]
miR-4524a-5p		High levels in patients with shorter time to first treatment		
miR-92a-3p	PBMCs ^1^	Lower levels in B-CLL patients vs. non-leukemic controls	Diagnosis	[98]
		Lower levels in patients with poor prognosis	Prognosis	
miR-155-5p	Plasma	Higher levels in B-CLL patients vs. non-leukemic controls	Diagnosis	[86]
	PBMCs ^1^	Higher levels in patients with poor prognosis	Prognosis	
	Purified B cells	Lower levels in responsive patients	Prediction of response	[85]
miRNA signature	Serum	-	Diagnosis	[99]
miRNA signature	-			[100]
miRNA signature	-		Prediction of progression	[89]
miRNA signature	PBMCs ^1^		Diagnosis; prognosis	[82]
miRNA signature	Purified B cells			[101]
miRNA signature	PBMCs ^1^		Prognosis; prediction of progression	[83]
Scoring system, including miR-21-5p	PBMCs ^1^	Higher levels in patients with poor prognosis	Prognosis	[91]
Scoring system, including miR-29c-3p		Lower levels in patients with poor prognosis		[90]

^1^ Peripheral blood mononuclear cells.

**Table 4 cancers-13-00593-t004:** miRNAs with a therapeutic interest with regard to B-CLL.

miRNA	Experimental Approach	Effect	References
miR-15a-5p; miR-16-5p	Human cells	Restoration of cell cycle control	[102]
	Mouse model	Drug sensitization and induction of apoptosis in mice upon its upregulation	[102,103,112]
miR-155-3p	Human cells	Regulation of chemoresistance	[104]
miR-222-3p	Human cells	Reduced cell viability and proliferation upon its downregulation	[105]
miR-181a-5p/miR-181b-5p	Human cells	Increased apoptosis of cells	[106,107]
	Mouse model	Reduced leukemic cell expansion and increase of survival in mice upon its upregulation	[107]
miR-34a-5p; miR-146b-5p	Human cells	Inhibition of cell proliferation upon its downregulation	[108]
miR-17-5p	Human cells	Reduced cell proliferation	[109]
	Mouse model	Reduced tumor growth and increased survival in mice upon its downregulation	
miR-26a-5p	Human cells	Induction of apoptosis with CD38-targeted delivery	[110]
	Mouse model		
miR-29b-3p	Mouse model	Induction of cell cycle arrest with ROR1-targeted delivery	[111]

## Data Availability

Not applicable.

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
