# Peer review of "MicroRNAs: Tiny Regulators of Gene Expression with Pivotal Roles in Normal B-Cell Development and B-Cell Chronic Lymphocytic Leukemia"

_cancers, 2021, doi:10.3390/cancers13040593_

Round 1
Reviewer 1 Report
Some refs are missing on the first two paragraphs in the Introduction, and it seems could merge them together.
Typo: line 54, stereotype
- In the introduction part, from line 45 to 61, authors mention some gene mutations in B-CLL but no reference is cited. And there are overlaps in these paragraphs, authors should consider to rewrite and reorganize the material.
- And one more suggestion, I don’t know if authors would like to add viral miRs to this review. Because some virus infection also play a important role in CLL progression, like EBV .
Author Response
Reviewer #1 (Comments to the Author):
1. Some refs are missing on the first two paragraphs in the Introduction, and it seems could merge them together.
Prompted by the Reviewer’s comment, we added the appropriate references in the Introduction part and merged the first two paragraphs.The new references are the following ones:
- Dohner, H.; Stilgenbauer, S.; Benner, A.; Leupolt, E.; Krober, A.; Bullinger, L.; Dohner, K.; Bentz, M.; Lichter, P. Genomic aberrations and survival in chronic lymphocytic leukemia. The New England journal of medicine 2000, 343, 1910-1916, doi:10.1056/NEJM200012283432602.
- International, C.L.L.I.P.I.w.g. An international prognostic index for patients with chronic lymphocytic leukaemia (CLL-IPI): a meta-analysis of individual patient data. Lancet Oncol 2016, 17, 779-790, doi:10.1016/S1470-2045(16)30029-8.
2. Typo: line 54, stereotype.
We fixed this issue.
3. In the introduction part, from line 45 to 61, authors mention some gene mutations in B-CLL but no reference is cited. And there are overlaps in these paragraphs, authors should consider to rewrite and reorganize the material.
According to the Reviewer’s comment, we modified the relative part by deleting repetitive information. Moreover, we added the two aforementioned reference.
4. And one more suggestion, I don’t know if authors would like to add viral miRs to this review. Because some virus infection also play an important role in CLL progression, like EBV.
We thank very much the Reviewer for this suggestion. We added in the revised manuscript a new section with title “3.4 Viral miRNAs in CLL” (lines 380-411), in which we discuss the current knowledge regarding the implication of EBV-derived miRNAs in CLL progression.
3.4. Viral miRNAs in B-CLL
Epstein-Barr virus (EBV) is a ubiquitous oncogenic human herpesvirus implicated in lymphomas, such as Burkitt’s lymphoma, while recent studies have associated EBV infection with Β-CLL progression [113]. More specifically, it was shown that Β-CLL patients with elevated levels of EBV DNA load had significantly shorter OS time intervals and characterized by therapy resistance, compared to the ones with lower levels of EBV DNA load [114,115]. However, further investigation is required regarding the potential mechanisms of EBV-driven oncogenesis in Β-CLL patients.
A proposed mode of action, via which EBV sustains viral infection, evades host immunity and potentially leads to oncogenesis, is based on viral miRNAs [116]. EBV was the first virus shown to encode viral miRNAs. To date, 44 mature miRNAs that could be classified into two clusters (BHRF1 cluster and BART cluster) have been identified in EBV. Particularly, the miRNAs from the BHRF1 cluster are expressed during lytic infection, inhibit apoptosis, and favor proliferation of infected cells to enable the early phase of viral propagation. Considering the role of these viral miRNAs, miR-BHRF1-1 has been further investigated in the context of Β-CLL. Precisely, the expression levels of miR-BHRF1-1 were significantly higher in the plasma of Β-CLL patients compared to the plasma of healthy individuals, while elevated levels of this miRNA were positively associated with advanced Rai stages. Furthermore, Β-CLL patients with elevated expression levels of miR-BHRF1-1 were characterized by shorter OS time intervals, compared to the ones with lower expression levels [117]. These findings designate the potential value of this miRNA as biomarker. Additionally, high expression levels of miR-BHRF1-1 were positively correlated with high miR-155-5p expression levels. This observation is quite significant, since miR-155-5p plays a decisive role in both normal B-cell development and Β-CLL progression, as it has been thoroughly analyzed above. The association between these two miRNAs has been also observed in another study, which showed that cellular miR-155-5p was induced by the viral miRNA and that miR-155-5p played a key role in B-cell immortalization [118]. Finally, infection of leukemic B cells with miR-BHRF1-1 reduces the levels of the key tumor suppressor, TP53 [117]. This interaction is, also, supported by an independent study, which suggested that this miRNA exerts its role in B-CLL via downregulation of TP53 and uncovered the therapeutic potential of miR-BHRF1-1 [119].
All these findings advocate the implication of EBV miRNAs in B-CLL onset and progression. Even though this research field is still in its infancy, the elucidation of the role of viral miRNAs in B-CLL progression is quite promising since it could further assist the discovery of novel biomarkers and therapeutic strategies.
We also added the appropriate references:
- Visco, C.; Falisi, E.; Young, K.H.; Pascarella, M.; Perbellini, O.; Carli, G.; Novella, E.; Rossi, D.; Giaretta, I.; Cavallini, C., et al. Epstein-Barr virus DNA load in chronic lymphocytic leukemia is an independent predictor of clinical course and survival. Oncotarget 2015, 6, 18653-18663, doi:10.18632/oncotarget.4418.
- Liang, J.H.; Gao, R.; Xia, Y.; Gale, R.P.; Chen, R.Z.; Yang, Y.Q.; Wang, L.; Qu, X.Y.; Qiu, H.R.; Cao, L., et al. Prognostic impact of Epstein-Barr virus (EBV)-DNA copy number at diagnosis in chronic lymphocytic leukemia. Oncotarget 2016, 7, 2135-2142, doi:10.18632/oncotarget.6281.
- Iizasa, H.; Kim, H.; Kartika, A.V.; Kanehiro, Y.; Yoshiyama, H. Role of Viral and Host microRNAs in Immune Regulation of Epstein-Barr Virus-Associated Diseases. Frontiers in immunology 2020, 11, 367, doi:10.3389/fimmu.2020.00367.
- Ferrajoli, A.; Ivan, C.; Ciccone, M.; Shimizu, M.; Kita, Y.; Ohtsuka, M.; D'Abundo, L.; Qiang, J.; Lerner, S.; Nouraee, N., et al. Epstein-Barr Virus MicroRNAs are Expressed in Patients with Chronic Lymphocytic Leukemia and Correlate with Overall Survival. EBioMedicine 2015, 2, 572-582, doi:10.1016/j.ebiom.2015.04.018.
- Linnstaedt, S.D.; Gottwein, E.; Skalsky, R.L.; Luftig, M.A.; Cullen, B.R. Virally induced cellular microRNA miR-155 plays a key role in B-cell immortalization by Epstein-Barr virus. Journal of virology 2010, 84, 11670-11678, doi:10.1128/JVI.01248-10.
- Xu, D.M.; Kong, Y.L.; Wang, L.; Zhu, H.Y.; Wu, J.Z.; Xia, Y.; Li, Y.; Qin, S.C.; Fan, L.; Li, J.Y., et al. EBV-miR-BHRF1-1 Targets p53 Gene: Potential Role in Epstein-Barr Virus Associated Chronic Lymphocytic Leukemia. Cancer research and treatment : official journal of Korean Cancer Association 2020, 52, 492-504, doi:10.4143/crt.2019.457.
The authors wish to thank the Reviewers for the constructive comments that led to the improvement of the current manuscript.

Reviewer 2 Report
This review seeks to outline the linkage between 37 miRNA function in normal and malignant hematopoiesis by demonstrating the main benchmarks 38 of the implication of miRNAs in the regulation of normal B-cell development, and to summarize the 39 key findings about their value as regulators, biomarkers, or therapeutic targets in B-CLL. The review is well written, is very rationally organized and provides all the essential data on this topic.
The modifications below suggested may improve this review paper:
- The authors should mention two recent studies on miRNA and CLL. A first study by Sharma et al. reports the role of miR-29 (miR-29a, -29b and -29c) downmodulated in CLL and involved in the modulation of CD40 signaling in CLL by targeting TRAF4 (Sharma et al. Blood 2020). A second study by Kour et al. was based on genome-wide small RNA sequencing and identified 3 miRNAs up-regulated and 5 miRNA down-regulated; some of these miRNAs have prognostic value (Kour et al., Blood Cancer J 2020, 10:6).
- The authors should better discuss the therapeutic implications related to some miRNA whose expression is deregulated in CLL.
Author Response
Reviewer #2 (Comments to the Author):
- The authors should mention two recent studies on miRNA and CLL. A first study by Sharma et al. reports the role of miR-29 (miR-29a, -29b and -29c) downmodulated in CLL and involved in the modulation of CD40 signaling in CLL by targeting TRAF4 (Sharma et al. Blood 2020). A second study by Kour et al. was based on genome-wide small RNA sequencing and identified 3 miRNAs up-regulated and 5 miRNA down-regulated; some of these miRNAs have prognostic value (Kour et al., Blood Cancer J 2020, 10:6).
We thank the Reviewer for this remark. In the revised manuscript, we mentioned both suggested studies (Table 2 and Table 3). Moreover, we included a sentence in the section “3.1. miRNAs as regulators in B-CLL” (lines 215-217): Furthermore, a recent study revealed the involvement of miR-29 family in CD40 signaling by targeting TRAF4, and proposed a novel regulatory axis in Β-CLL, modulated by the BCR activity [62].
We also added the appropriate references:
- Sharma, S.; Pavlasova, G.M.; Seda, V.; Cerna, K.A.; Vojackova, E.; Filip, D.; Ondrisova, L.; Sandova, V.; Kostalova, L.; Zeni, P.F., et al. miR-29 Modulates CD40 Signaling in Chronic Lymphocytic Leukemia by Targeting TRAF4: an Axis Affected by BCR inhibitors. Blood 2020, 10.1182/blood.2020005627, doi:10.1182/blood.2020005627.
- Kaur, G.; Ruhela, V.; Rani, L.; Gupta, A.; Sriram, K.; Gogia, A.; Sharma, A.; Kumar, L.; Gupta, R. RNA-Seq profiling of deregulated miRs in CLL and their impact on clinical outcome. Blood cancer journal 2020, 10, 6, doi:10.1038/s41408-019-0272-y.
- The authors should better discuss the therapeutic implications related to some miRNA whose expression is deregulated in CLL.
Taking into consideration the Reviewer’s comment, we changed the last sentence in section “3.3. miRNAs in B-CLL therapy” (lines 373-376), as presented here below: However, prior to all these ameliorations, it is important to clarify the miRNA regulatory-disease specific network of miRNAs with deregulated levels in B-CLL, in order to distinguish the most promising miRNAs for treatment, and therefore achieve a miRNA-based therapy. Moreover, we included a small part in section “4. Future perspectives” (lines 4510-454): Increasing evidence shows specific miRNAs with deregulated levels in Β-CLL, in comparison with normal individuals, suggesting miRNAs as biomarkers. Moreover, the alteration of their levels is a tightly regulated and finely tuned process. Therefore, elucidating the regulatory effect of miRNAs with deregulated levels may reveal other miRNAs with therapeutic potency.
The authors wish to thank the Reviewers for the constructive comments that led to the improvement of the current manuscript.
